# Wavelet-Based Contourlet Transform and Kurtosis Map for Infrared Small Target Detection in Complex Background

**DOI:** 10.3390/s20030755

**Published:** 2020-01-30

**Authors:** He Wang, Yunhong Xin

**Affiliations:** School of Physics and Information Technology, Shaanxi Normal University, Xi’an 710119, China; _hwang@snnu.edu.cn

**Keywords:** infrared small target detection, Wavelet-based Contourlet transform (WBCT), kurtosis map, complex background

## Abstract

Wavelet-based Contourlet transform (WBCT) is a typical Multi-scale Geometric Analysis (MGA) method, it is a powerful technique to suppress background and enhance the edge of target. However, in the small target detection with the complex background, WBCT always lead to a high false alarm rate. In this paper, we present an efficient and robust method which utilizes WBCT method in conjunction with kurtosis model for the infrared small target detection in complex background. We mainly made two contributions. The first, WBCT method is introduced as a preprocessing step, and meanwhile we present an adaptive threshold selection strategy for the selection of WBCT coefficients of different scales and different directions, as a result, the most background clutters are suppressed in this stage. The second, a kurtosis saliency map is obtained by using a local kurtosis operator. In the kurtosis saliency map, a slide window and its corresponding mean and variance is defined to locate the area where target exists, and subsequently an adaptive threshold segment mechanism is utilized to pick out the small target from the selected area. Extensive experimental results demonstrate that, compared with the contrast methods, the proposed method can achieve satisfactory performance, and it is superior in detection rate, false alarm rate and ROC curve especially in complex background.

## 1. Introduction

Infrared small target detection has been a hot topic for guidance, defense, navigation, infrared search and track and other photoelectric imaging systems [1,2,3,4]. In the past few decades several classical methods have been proposed, such as max-mean filter and max-median filter [5], Two-Dimensional Least Mean Square [6], bilateral filter [7], morphological filter [8]. However, these baseline methods fail to obtain satisfactory performance in complex background with low SNR.

In recent years, multiscale geometric analysis (MGA) has shown substantial advantages in target detection area. MGA makes up for the shortcomings of wavelet transform [9] in capturing direction information, its representative methods include ridgelet transform [10], curvelet transform [11], contourlet transform [12], and shearlet transform [13]. The Contourlet transform can make the most of the geometric characteristics of data, such as line singularities and plane singularities. Since contourlet transform contains the downsampling, it lacks shift invariant property [14]. Researchers have proposed Nonsubsampled contourlet transform (NSCT) to describe complex spatial structures in various directions well. Both the Contourlet transform and the NSCT transform are implemented based on the Laplacian pyramid. Due to the redundancy of the Laplacian pyramid, both of these transforms have high redundancy factor. Eslami and Radha [15] proposed a new non-redundant image transform which named Wavelet-based Contourlet transform. In target detection, MGA can not only accurately capture the singularity of an image, but also suppress the background noise and highlight the target. Up to now, there are many MGA type methods have been applied to infrared small target detection, and achieve good detection results. For example, Qu et al. [16] proposed a small target detection method based on curvelet neural network to improve the recognition rate of the complex background. Zhao et al. [17] proposed a new detection method based on nonsubsampled contourlet transform (NSCT) to achieve high detection rate in low SNR infrared image. Ji et al. [18] used contourlet transform combined with principal component analysis (PCA) to achieve strong anti-noise performance and target detection performance. Lei et al. [19] adopted an optimized fast NSCT to maintain better time-consuming performance for small target detection. Peng et al. [20] introduced the Shearlet and maximum kurtosis to infrared target detection and obtain good performance. The method Gaussian Scale-Space and Spectral Scale-Space belong to another multiscale method. The image is decomposed into multiscale sub-bands by these methods, and small target is detected by processing multiscale sub-bands. Xin et al. [21] proposed a small target detection method based on Spectral Scale-Space and Gabor Wavelet to gain high correct target detection rate. Yi et al. [22] used an improved spectral scale space combined with the Adaptive Local Contrast Measure method to small target detection, which improved the efficiency of small target detection under complex backgrounds. Guan et al. [23] presented a fast and effective detection method based on Gaussian scale space and Enhanced Local Contrast Measure. The multi-scale method can enhance targets by eliminating the noise and background, but also enhance the background edges.

Recently, methods based on human visual system (HVS) have shown many advantages in the field of target detection. HVS generates a saliency map to identify the regions of interest (ROI) and detects target in ROI. Guo et al. [24] calculated the spatiotemporal saliency map based on phase spectrum of quaternion Fourier transform to against white-colored noise in target detection. Wang et al. [25] performed the analysis of small target grey intensity surface to establish an efficient and reliable small target detection method based on facet model. Chen et al. [26] processed a new contrast measuring method named local contrast measure (LCM). This method generates a saliency detection map by using the dissimilarity between the target and the surrounding background. Chen et al. [27] combined local self-similarity with local contrast to compute the local saliency map for small target detection, from which the target region can be highlighted. Deng et al. [28] proposed a multiscale gray difference weighted image entropy small target detection method, which effectively enhanced small target. These HVS-based methods are usually superior to baseline methods in detection performance, However, some of them cannot detect the target accurately when apply to the images with highly heterogeneous backgrounds. 

In this paper, an efficient and robust small target detection method in single-frame infrared image is presented. The presented method consists of a preprocessing stage and a detection stage. At the preprocessing stage, WBCT is utilized to decompose the original infrared image into multiscale and multidirectional sub-bands, and then an adaptive threshold selection strategy is applied to select the background information in high frequency sub-bands, and the coarse target image is obtained by making a difference between the original image and the background image after the Inverse Wavelet-based Contourlet transform (IWBCT) transformation. In this stage, most background clutters are suppressed and the target is enhanced. At the detection stage, an effective kurtosis map calculated by local kurtosis operator combined with the adaptive threshold segmentation mechanism is applied to process the coarse target image, by which the residual background edge is efficiently eliminated. Extensive experimental results demonstrate that the proposed method can achieve satisfactory performance.

## 2. Related Work 

### 2.1. Wavelet-Based Contourlet Transform (WBCT)

The Contourlet transform consists of a Laplacian Pyramid (LP) and a Directional Filter Bank (DFB). The WBCT replaces the LP in the Contourlet transform with a non-redundant Mallat tower. Wavelet transform firstly implements the multiscale decomposition, and then DFB implements the angular decomposition. Figure 1 illustrates the implementation principle of WBCT. 

WBCT has the advantages of Wavelet transform and Contourlet transform, which inherits the multi-resolution of wavelet transform, and overcomes the problem of the redundancy of the Contourlet transform. In this paper, the image is made 3 level wavelet transform decomposition. Then the high-pass sub-bands ($LH_{1},\;HL_{1},\;HH_{1}$) of the first level are decomposed in 8 directions, the high-pass sub-bands ($LH_{2},\;HL_{2},\;HH_{2}$) of the second level are decomposed in 4 directions, the high-pass sub-bands ($LH_{3},\;HL_{3},\;HH_{3}$) of the third and the low-pass sub-band ($LL_{3}$) level are not decomposed. This frame is in favor of background suppression. 

### 2.2. Kurtosis Model

The kurtosis is used to measure the degree of convergence of random signals in the center [29]. The kurtosis is defined as: (1)$$K=\frac{M_{4}}{M_{2}^{2}}-3$$
where $M_{4}$ represents the 4th-order sample central moment, $M_{2}$ represents the 2th-order sample central moment.

If the value of the kurtosis is 0, it means that the random variable is the same as the normal distribution. If it is greater than 0, it means that the random variable is more concentrated, and there is a tail shorter than the normal distribution. If it is less than 0, the random variable is not concentrated, and there is a tail that is longer than the normal distribution. As shown in Figure 2:

## 3. Proposed Method

The frame of the proposed Infrared image small target detection method is shown as Figure 3. It can be seen that the proposed method is divided into two stages: the preprocessing stage and the detection stage. At preprocessing stage, the input image is decomposed into the multiscale and the multidirectional sub-bands by WBCT, and then an adaptive threshold selection strategy is applied to the high frequency sub-band, which could remove small targets information and remain background information. Next, the background image is obtained By IWBCT of the processed high frequency sub-bands and low frequency sub-band. After then, a coarse target image is obtained by a subtract operation of the input with the background image. At detection stage, the kurtosis map is obtained by local kurtosis operator of the coarse target image, and then a sliding window and its associated mean and variance are used to determine the small target area in kurtosis map. Finally, an adaptive threshold segmentation mechanism is applied to extract the small target from the selected area. 

### 3.1. Preprocessing Stage

In this part, we adopt an adaptive threshold selection strategy to remove detail information from WBCT sub-band coefficients and recover the background information. The image f(i,j) could be represented mathematically as:(2)f(i,j)=b(i,j)+t(i,j)
where b(i,j) represents the background of the image, t(i,j) represents the detail of the image, such as small target or edge of the object.

After the infrared image f(i,j) is decomposed by WBCT, it can be represented as:(3)$$C_{i,j}=S_{i,j}+N_{i,j}$$
where $C_{i,j}$, $S_{i,j}$, and $N_{i.j}$ represent sub-band coefficients of the original image f(i,j), the background information b(i,j) and the detail information t(i,j) after WBCT decomposed. The background information b(i,j) is mainly in low frequency sub-band, while the detail information t(i,j) is mainly in high frequency sub-bands. Thus, the focus of this stage is to separate background information and detail information in high frequency sub-bands. According to the wavelet domain denoising model [30], an effective soft-thresholding function is applied to remove the detailed information in high frequency sub-bands, represented as:(4)$${\hat{C}}_{i,j}=\;\{\begin{array}{c} C_{i,j}-T,\;\;\;\;\;\;\;\;\;\;\;\;C_{i,j}>T\;\\ C_{i,j}+T,\;\;\;\;\;\;\;\;\;\;C_{i,j}<-T\\ \;\;\;0,\;\;\;\;\;\;\;\;\;\;\;\;\;\;\;-T\leq C_{i,j}\leq T\; \end{array}$$
where T represents threshold, the optimal threshold *T* is defined to be the argument which minimizes the expected squared error, it can be well approximated by
(5)$$T=\frac{{\hat{\sigma }}_{N}^{2}}{{\hat{\sigma }}_{S}}$$
where ${\hat{\sigma }}_{N}^{2}$ represent the variance estimation of detail information, ${\hat{\sigma }}_{S}$ represent the variance estimation of background information. 

According to Equation (3), sub-band coefficients are hybrid matrix of the background information and the detail information, so the variance of $C_{i,j}$ is also viewed as the variance of $S_{i,j}$ and $N_{i,j}$. Here, we assume that the background information b(i,j) and the detail information t(i,j) are independent of each other, so the variance $\sigma_{S}^{2}$ of background information can be represented as:(6)$$\sigma_{S}^{2}=\sigma_{C}^{2}-\sigma_{N}^{2}$$
where $\sigma_{C}^{2}$ is the variance of sub-band coefficients after WBCT decomposition. The variance $\sigma_{N}^{2}$ of the detail information is estimated by using the robust median estimator in the highest sub-band of the wavelet transform:(7)$${\hat{\sigma }}_{N}^{2}=\frac{median\left(|C_{i,j}|\right)}{0.6745},\;C_{i,j}\in HH_{1}$$
which is also used in [30]. By estimating the parameter $\sigma_{S}^{2}$ for each sub-band, we can get a uniform threshold T. However, using a single threshold for sub-bands of different scales, frequencies and orientations is lack of credibility. Considering the continuity of background information in frequency sub-band, one has the luxury of estimating the parameter $\sigma_{S}^{2}$ for each coefficient by using a moving window. For a given coefficient $C_{i_{0}j_{0}}$, its variance is estimated $\sigma_{S}^{2}\left(i_{0},j_{0}\right)$ is represented as:(8)$$\sigma_{S}^{2}\left(i_{0},j_{0}\right)=max\left\{\frac{1}{N_{\beta }}\underset{\left(i,j\right)\in \beta_{i,j}}{\sum }C_{i,j}{}^{2}-\sigma_{N}^{2},0\right\}$$
where the set β denote the pixels contained in the window centered on point ($i_{0},j_{0}$). The $N_{\beta }$ represent the number of coefficient contained in set β. Then calculating the threshold $T_{i,j}$ for every location (*i*,*j*) according to Equations (5)–(8) yields an adaptive threshold. The process of the preprocessing stage is described as follows:(1)The infrared image is decomposed into multiscale and multidirectional sub-bands as shown in Figure 1.(2)In each high frequency sub-band, set a sliding window to traverse the entire sub-band from left and top to right and down. In the sliding window, the threshold of the current window center coefficient is estimated by using Equations (5)–(8).(3)According to the spatially adaptive threshold, the soft-thresholding function is used to remove the detailed information in each high frequency sub-band respectively.(4)Inverse transform of WBCT is performed on the low frequency sub-band and the processed high frequency sub-band coefficients, and difference is made with the original image to obtain the coarse target image.

Considering that the sub-band of each level obtained after the image is decomposed by WBCT is smaller than the previous level, so the size of the sliding window is used in each level is also smaller than the previous level. Here, the sliding window size set as 9×9, 7×7 and 3×3 in the high frequency sub-band of first, second and third level. 

Figure 4 shows the preprocessed results of different images. The upper left image in the second row images is their three-dimensional diagrams. It can be seen when there are few clouds or buildings in the image, the background of the infrared image is effectively suppressed, and small targets can be detected by simple threshold segmentation (as shown in Figure 4(b1,b2)). However, when there are “bright” man-made buildings or trees in the image, some edges of the infrared image are not completely suppressed. If simple threshold segmentation is performed on these images, the residual edges will cause false detection (as shown in Figure 4(b3,b4)). Therefore, detecting the small target accurately in the residual background edge is a critical stage.

### 3.2. Detection Stage

In this part, we introduce a method based on kurtosis map how to eliminate residual background edge. This method uses a local kurtosis operator to generate a significant kurtosis map. In the kurtosis map, the area where the small target could be effectively located, and then adaptive threshold segmentation is used for small target detection. It is good improving small target detection in infrared image.

#### 3.2.1. Local Kurtosis Operator

The kurtosis reflects the difference between the sample distribution and the normal distribution. For an image, if its gray values have a homogeneous distribution its kurtosis value would be small, on the contrary if have a inhomogeneous distribution its kurtosis value would be large. Generally, the kurtosis value is greater in small target image than in background image. However calculating kurtosis values for the entire image neglects the texture and frequency information of an image. Thus, the local kurtosis operator is restricted in a local window whose size is m×n, and it can represent the information content contained in the window. When a small target appears in an infrared image, it will intensify the change of the gray value of the local area around the target, thereby making the kurtosis value of the local area larger. In a sense, the local kurtosis operator can be used to enhance small infrared targets under complex backgrounds. The local kurtosis operator can be expressed as:(9)$$K=\frac{1}{N_{\Omega }}\frac{\sum_{\left(x,y\right)\in \Omega }{\left(f\left(x,y\right)-\mu \right)}^{4}}{\sigma^{4}}-3$$
where Ω denote the pixels contained in local window whose size as m×n, $N_{\Omega }$ is the number of the pixels contained in set Ω. 

#### 3.2.2. Kurtosis Map

The preprocessed result is shown in Figure 4b, indicating that the background is well suppressed. However, for infrared image against complex terrain-sky or sky backgrounds, the edge of the object has similar thermal intensity measure as a small target which greatly affects the performance of the WBCT method. Therefore, a kurtosis map is calculated by local kurtosis operator to offset this effect, improving the adaptability of the WBCT method. The method to calculate the kurtosis map as shown in Algorithm 1, where m×n is the size of the neighboring area centered at the pixel point (*i*,*j*), and m and n are odd integet numbers, respectively. The kurtosis map is shown in Figure 5.
**Algorithm 1** Calculating the Kurtosis map**Input:** Given an input image f of size M×N
**Output:** The Kurtosis map *K*(1)**for** *i* = 1 : *M*
**do**(2)**for** *j* = 1 : *N*
**do**
$$K\left(i,j\right)=\frac{1}{mn}\frac{\sum_{x=i-\left(m-1\right)/2}^{x=i+\left(m-1\right)/2}\sum_{y=j-\left(n-1\right)/2}^{y=j+\left(n-1\right)/2}{\left(f\left(x,y\right)-\mu \right)}^{4}}{\sigma^{4}}-3$$  Replace the value of the center pixel with the K(*i*,*j*).(3)**end for**(4)**end for**

In Figure 5, a small white patch (marked by the green circle in the figure) appears in the target area. This shows that when calculating the kurtosis of a small target and its neighboring image patch, the kurtosis value of these patch affected by the small target is generally large. However, the kurtosis value of the background edge is larger in a certain direction. In addition, experiments demonstrate that when different image patch size is used to calculate the kurtosis map, the size of small white patch is different. The larger image patch is used to calculate the kurtosis map, the larger small white patch is obtained. At the same time, the difference between the kurtosis characteristic of the small target area and the background edge area is larger. But when the image patch is increased to a certain extent, the difference will be no longer obviously. Therefore, the chosen image patch size will greatly affect the detection results of the proposed method.

#### 3.2.3. Area Location

According to the kurtosis map, the small target and the background edge show different characteristics. The specific method for determining the area where the small target is located as follows:(1)Calculate the kurtosis map according to Algorithm 1. Experimental data shows that this we select the image patch size as 13×13 have a better effect than other size.(2)Determine the sliding window size m×n. Considering the size of small target and the calculate speed of method, we still select the window size as 13×13.(3)Set the sliding step (we set it as the half of the sliding window). Calculate the mean and variance within each sliding window as the sliding window sliding from left and top to right and down in the kurtosis map, denoted as $\mathrm{mean}{\left(K\right)}_{i}$ and $D{\left(K\right)}_{i}$.(4)Set the threshold T. Based on the sensitivity study, let $T=\alpha \times \mathrm{mean}{\left(K\right)}_{\max},\;\alpha \epsilon \left[0.8,1\right]$. If $\mathrm{mean}{\left(K\right)}_{i}$ > T, it is considered that the small target may be in the ith image patch, otherwise it is considered as background clutter and discarded.(5)The image patch with the smallest variance among the image patches whose mean value is greater than *T* is considered to be the small target area, and the rest are the background edge areas and removed.


#### 3.2.4. Target Segmentation

In the previous step, the small target area is located. In this step small target is segmented by adaptive threshold in the area. The adaptive threshold Th is defined as:(10)Th=μ+k×δ
where μ and δ are the mean and standard deviation of the image patch in the preprocessed image respectively. The parameter *k* is an empirical value from experiments, usually choosing 1 in our experiments. The pixels whose gray value is larger than Th are identified as target and the remaining pixels are background and discarded.

## 4. Experiments and Analysis

In our experiments, 8 real small target image sequences with low SCR values (denotes as Sequence 1 to Sequence 8) are utilized to test the performance of the proposed method. The real IR images are taken by the members of our lab with IR detector outdoors. The most of test sequences have low contrast. The resolution of the images is 288 × 384. The details about targets and backgrounds are listed in Table 1.

To verify the validity of the proposed method, we compared it with state-of-the-art multiscale methods, e.g., DWT, Spectrum Scale-Space and Gabor wavelet (SSS-GW) [19], Fast Spectrum Scale-Space and Adaptive Local Contrast Measure (TDGS is used for shorthand notation for method in paper [20]) and Gaussian Scale-Space and Enhanced Local Contrast Measure (GSS-ELCM) [21] as four baseline methods. Moreover, Novel Weighted Image Entropy (NWIE) [26] based on Local Contrast Measure method is also chosen as the baseline method in this paper. All the experiments were conducted on a computer with 8-GB RAM and 3.6-GHz Intel core i7CPU, and the code was implemented in MATLAB R2017a.

Figure 6 illustrates the test images and detection results of different methods, it can be seen that the proposed method can effectively suppress complex backgrounds and all targets are accurately detected without missing detection in eight sequences. The method NWIE produces a strong response at the most background. When the image contains a little “bright” buildings or trees, the saliency maps of the NWIE also enhance the object edges, and the false alarms emerge in Figure 6(b6,b7). The method DWT can suppress the cloudy background and detect the target, such as Figure 6(c1,c3,c5). But the DWT cannot suppress the clutter, making the target drowned in noise, such as Figure 6(c2,c6,c8). The method SSS-GW can suppress background and most noise. In Figure 6(d1–d5), it can detect the target correctly. However, the SSS-GW cannot make the target isolated from the interferences in Figure 6(d7,d8). The method TDGS can detect the target in Figure 6(e1,e3,e5). However, the TDGS have the similar results with DWT in some complex images. The method GSS-ELCM shows a good performance in most images. But when the target is very dim (possibly much lower brightness than the background), the GSS-ELCM can only detect a few pixels of small target, such as Figure 6g8, however the other baseline methods can lead to real target missed.

In order to further illustrate the superiority of our method, this paper introduces two evaluation indicators: signal-to-clutter ratio gain (SCRG) and background suppression factor (BSF). They are defined as:(11)$$\mathrm{SCR}=\frac{\mu_{t}-\mu_{b}}{\sigma_{b}}$$
(12)$$\mathrm{SCRG}=20\times \log_{10}\left(\frac{SCR_{\mathrm{out}}}{{\mathrm{SCR}}_{\mathrm{in}}}\right)$$
(13)$$\mathrm{BSF}=\;\frac{\sigma_{in}}{\sigma_{out}}$$
where $\mu_{t}$, $\mu_{b}$ represent the gray mean of the small target and the image background, $\sigma_{b}$ represents the standard deviation of the image background; $\sigma_{in}$ and $\sigma_{out}$ represents the standard deviations of the input image and the output image.

The SCRG and BSF of the six methods for eight image sequences are respectively shown in Table 2 and Table 3, where $\bar{\mathrm{SCRG}}$ and $\bar{\mathrm{BSF}}$ are the average SCRG and BSF in five frame images randomly selected from the eight real image sequences, respectively. The larger SCRG value means a larger gray difference between the small target and the background. The larger BSF value means stronger the suppression of the background. It can be seen that our method have the highest values among the comprised methods which means our method obtain the best performance in background suppression and small target extraction.

The receiver operating characteristic (ROC) curve, another metric often used to evaluate the effectiveness of the method, is used in our experiment. The ROC curve describes the relationship between the probability of detection (Pd) and the false alarm rate (Fa). The definitions of Pd and Fa are as follows:(14)$$\mathrm{Pd}\;=\;\frac{\mathrm{number}\;\mathrm{of}\;\mathrm{detected}\;\mathrm{true}\;\mathrm{targets}}{\mathrm{total}\;\mathrm{number}\;\mathrm{of}\;\mathrm{real}\;\mathrm{targets}}$$
(15)$$\mathrm{Fa}\;=\;\frac{\mathrm{number}\;\mathrm{of}\;\mathrm{detected}\;\mathrm{false}\;\mathrm{targets}}{\mathrm{total}\;\mathrm{number}\;\mathrm{of}\;\mathrm{pixels}\;\mathrm{in}\;\mathrm{the}\;\mathrm{whole}\;\mathrm{image}}$$

Figure 7 shows the ROC curves of six methods for eight real images. Our method has better performance than baseline methods and possesses higher Pd but lower Fa, compared with the baseline methods, especially for Sequence 6, 7, and 8. For the Sequence 5, the SSS-GW and TDGS are slightly superior to the proposed method when Fa > 0.7 or Fa > 0.84. In general terms, the ROC curves demonstrate that the proposed method is robust and appropriate to detect small targets against complex backgrounds.

## 5. Conclusions

In this paper, we present an efficient and robust infrared small target detection method based on WBCT and kurtosis map. Some representative methods such as NWEI, DWT, SSS-GW, TDGS and GSS-ELCM are used as contrast methods to demonstrate the performance of the presented method. In the experiment, we have processed eight typical scenarios such as cloud-sky background, terrain-sky background, “bright” buildings background, heavy noise, and their hybrid background. The experimental result image and corresponding 3D diagrams show that the proposed method can well suppress the background and accurately detect small targets under a variety of backgrounds. The BSF value and SCRG value of the proposed method are also larger than the baseline methods. The comparisons derived from the ROC curves demonstrate that the proposed method has a high detection rate, except for a few special cases which the Pd value of the same Fa is lower than the contrast method. In general, it is an efficient method to the IR dim and small target detection in the complex background. 

## Figures and Tables

**Figure 1 sensors-20-00755-f001:**
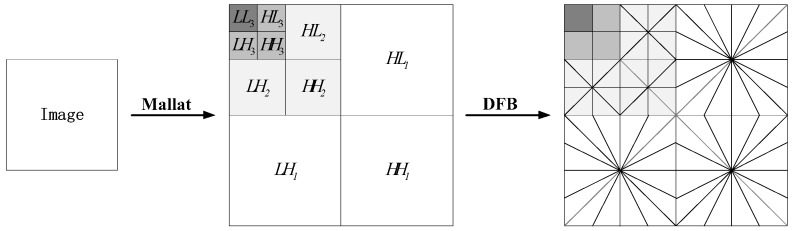
Flow chart of WBCT implementation principle.

**Figure 2 sensors-20-00755-f002:**
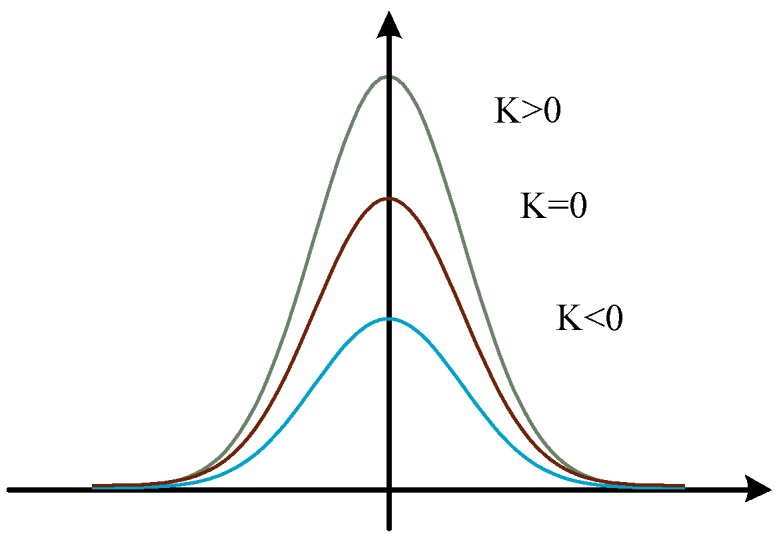
Illustration of sub-Gaussian, sup-Gaussian, and Gaussian distributions. Their Kurtosis value are also presented.

**Figure 3 sensors-20-00755-f003:**
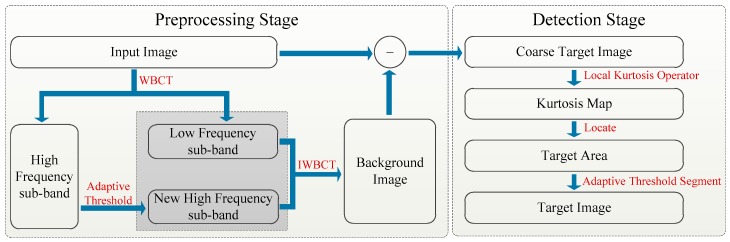
The frame of the proposed Infrared image small target detection method.

**Figure 4 sensors-20-00755-f004:**
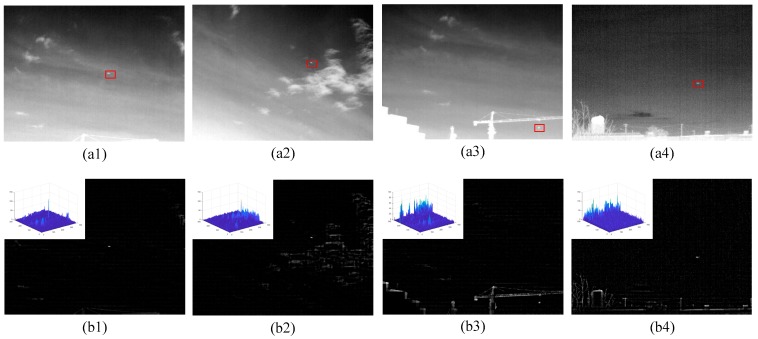
The preprocessed results of different images. (**a**) Original image, (**b**) Image after preprocessing stage.

**Figure 5 sensors-20-00755-f005:**
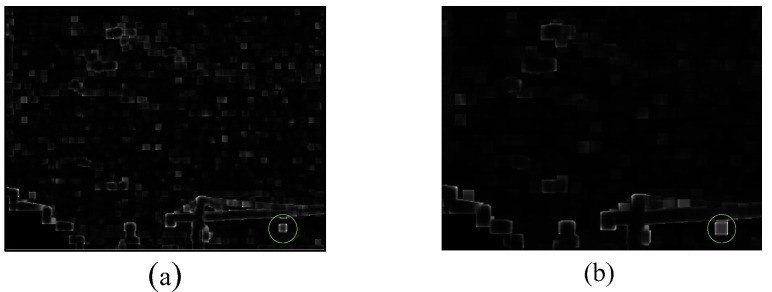
The kurtosis map of calculated with respect to different patch sizes (**a**) the patch size is 7 × 7, (**b**) the patch size is 13 × 13.

**Figure 6 sensors-20-00755-f006:**
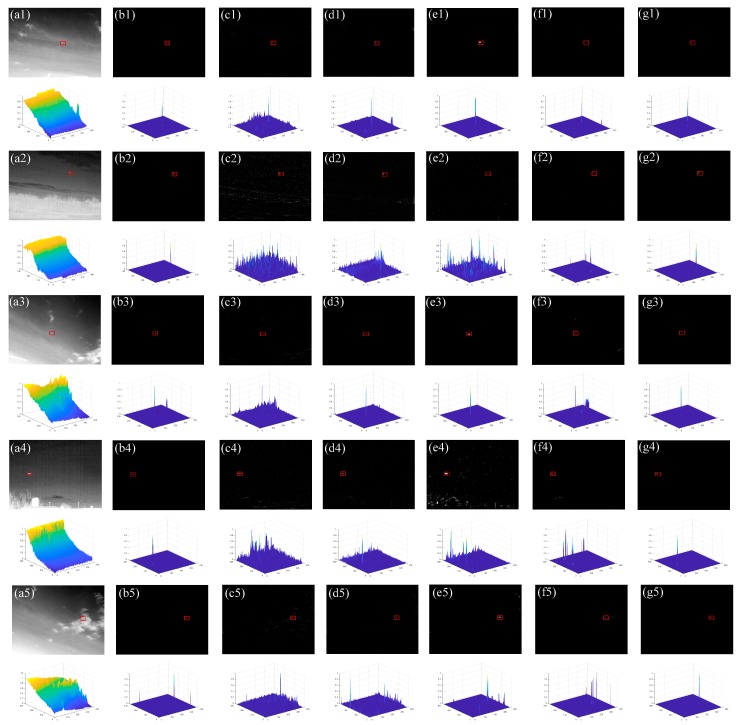
Test images and detection results of different methods. (**a1**–**a8**) test images and corresponding 3D diagrams of Sequence 1 to Sequence 8, respectively, (**b1**–**b8**) processed images of method NWIE and corresponding 3D diagrams, (**c1**–**c8**) processed image of methods DWT and corresponding 3D diagrams, (**d1**–**d8**) processed images of method SSS-GW and corresponding 3D diagrams, (**e1**–**e8**) processed images of method TDGS and corresponding 3D diagrams, (**f1**–**f8**) processed images of method GSS-ELCM and corresponding 3D diagrams, (**g1**–**g8**) processed images of our method and corresponding 3D diagrams. Red rectangle marks the location of the real small target.

**Figure 7 sensors-20-00755-f007:**
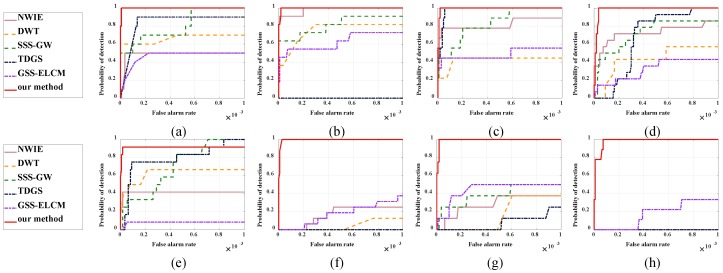
ROC curves of different methods for eight real images. (**a**) Sequence 1, (**b**) Sequence 2, (**c**) Sequence 3, (**d**) Sequence 4, (**e**) Sequence 5, (**f**) Sequence 6, (**g**) Sequence 7, (**h**) Sequence 8.

**Table 1 sensors-20-00755-t001:** The details of the eight real IR sequences.

Sequence	Target Size	Target Details	Background Details
Sequence 1	2×5	A small size with a little change. Keeping motionless.	Cloud-sky backgrounds. Uniform backgrounds.
Sequence 2	3×4	Low SNR value. Keeping motion. A dim target within a big range.	Terrain-sky background. Heavy noise. Almost keeping the same.
Sequence 3	2×4	A long imaging distance. A small size. Keeping motion.	Terrain-sky background. Changing background.
Sequence 4	3×6	Low SNR value. A changing size within a big range.	There are “bright” buildings. Almost keeping the same. Heavy noise.
Sequence 5	3×4	A small size. Hidden in the clouds. Keeping motion.	Cloud-sky background. Changing backgrounds. Heavy cloud clutter.
Sequence 6	2×4	A dim target. Keeping motionless. Low SNR value.	There are “bright” buildings. Almost keeping the same. Heavy noise.
Sequence 7	3×4	A long imaging distance. A small size. Keeping motion.	There are “bright” buildings. Changing backgrounds.
Sequence 8	3×3	A dim target with a little change. Low SNR value.	Heavy noise. Uniform backgrounds. There are “bright” buildings.

**Table 2 sensors-20-00755-t002:** The Values of $\bar{\mathrm{SCRG}}$ of Results Obtained Different Methods.

	Sequence 1	Sequence 2	Sequence 3	Sequence 4	Sequence 5	Sequence 6	Sequence 7	Sequence 8
NWIE	33.5057	64.2209	32.6168	24.3036	29.1318	23.0811	17.9498	18.8251
DWT	29.0982	49.8862	24.0603	15.4281	23.7058	15.1744	11.9642	12.5335
SSS-GW	33.9181	61.2395	32.6128	22.4076	34.8627	12.1550	17.5429	17.5581
TDGS	34.2707	46.3169	34.5819	19.8412	34.2712	14.3744	−3.2065	3.6179
GSS-ELCM	30.8430	64.0019	27.2701	17.6308	9.6349	21.3876	21.1241	20.1698
Our method	**37.2318**	**70.5383**	**38.5675**	**31.8643**	**40.2450**	**39.3232**	**34.9650**	**44.2755**

**Table 3 sensors-20-00755-t003:** The Values of $\bar{\mathrm{BSF}}$ of Results Obtained Different Methods.

	Sequence 1	Sequence 2	Sequence 3	Sequence 4	Sequence 5	Sequence 6	Sequence 7	Sequence 8
NWIE	30.9016	23.5972	27.3800	25.2920	23.2550	16.7425	18.7937	21.1644
DWT	24.3641	13.5543	19.7854	17.3074	21.1133	16.5301	15.5757	14.9175
SSS-GW	29.6892	22.9971	30.2402	24.4064	31.4539	13.6742	21.0282	19.5508
TDGS	29.0140	18.4452	30.3124	21.4572	28.7883	22.2547	25.4452	22.4326
GSS-ELCM	29.1536	26.6688	25.2412	23.0235	19.8636	23.6987	26.6781	27.2831
Our method	**32.3398**	**29.5252**	**32.3920**	**29.2573**	**32.7446**	**27.8692**	**30.1006**	**29.6718**

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
