# Peer review of "Wavelet-Based Contourlet Transform and Kurtosis Map for Infrared Small Target Detection in Complex Background"

_sensors, 2020, doi:10.3390/s20030755_

Round 1

Reviewer 1 Report

Dear authors,
you wrote a valuable article. My suggestion is you can give more space for the conclusion, where you can summarize all results with keeping the focus on applications of your proposed method.
In the text, I could not find the explanation for shortcut IWBCT. I suppose it is Inverse WBCT. However, I suggest to write it explicitly at the place where shortcut IWBCT appears for the first time in the text.

Author Response

We have responded to the reviewers' comments point-by-point, and the detail content has been uploaded as a PDF file.

Reviewer 2 Report

This is an interesting research paper. There are some suggestions for revision.

1. The motivation is not clear. Please explain the importance of the proposed solution.

2. Please highlight the contributions of the proposed solution in introduction.

3. In related work, it discusses WBCT and kurtosis model. But the discussion is too general. Please specify the connection between WBCT/kurtosis
model and the proposed solution. Existing related solutions should be discussed and their pros and cons should also be compared.

4. Please discuss the following Laplacian decomposition solution.
Y. Li, Y. Sun, X. Huang, G. Qi, M. Zheng, Z. Zhu, An Image Fusion Method Based on Sparse Representation and Sum Modified-Laplacian in NSCT Domain, Entropy 20(7): 522 (2018)

5. As shown in Eq. 6, what is delta c? It is not clear. How to get the value of delta c?

6 Where is Eq. 7 from, including median and 0.6745?

7. Please add more steps to show how to get Eq. 8 and explain why it needs to get the maximum value.

8. On page 5, it mentions the size of the sliding window. Please explain how to choose the suitable size of the sliding window.

9. The top left figures in Fig. 4 b1-b4 are too small. Please add more explanations.

10. Where is Eq. 9 from? How to choose the local window size m*n? What is the value of mu.

11. In Algo. 1, it should add indentation between step 2 and step 5.

12. Please show where the images used in the experiment come from.

13. The best results in Tab. 2 and 3 should be highlighted.

14. The sub-figs in Fig. 6 are too small to figure out the difference. Please increase the figure size and add more explanations.

Author Response

(The authors gave the same response as above.)

Round 2

Reviewer 2 Report

All my concerns have been addressed. This paper is ready for publication.